Clinical significance of small nuclear ribonucleoprotein U1 subunit 70 in patients with hepatocellular carcinoma

Jiang Dong 1
Zhu Xia-Ling 1
An Yan anyan0508@163.com 2
Li Yi-ran liyiranehsh@sina.com 3
1 Department of Ultrasound, Eastern Hepatobiliary Surgery Hospital, The Third Affiliated Hospital of Naval Medical University , Shanghai , China
2 Hongqiao International Institute of Medicine, Tongren Hospital, Shanghai Jiao Tong University School of Medicine , Shanghai , China
3 Department of Intensive Care Medicine, Eastern Hepatobiliary Surgery Hospital, The Third Affiliated Hospital of Naval Medical University , Shanghai , China
Adnan Mohd
Electronic publication date: 2024 Mar 15
Publication date: 2024
Volume: 12
Electronic Location ID: e16876
Received 2023 Apr 7; Accepted 2024 Jan 11
Copyright: ©2024 Jiang et al.
Copyright year: 2024
Copyright holder: Jiang et al.
License: This is an open access article distributed under the terms of the Creative Commons Attribution License, which permits unrestricted use, distribution, reproduction and adaptation in any medium and for any purpose provided that it is properly attributed. For attribution, the original author(s), title, publication source (PeerJ) and either DOI or URL of the article must be cited.
License URL: https://creativecommons.org/licenses/by/4.0/

Keywords: Hepatocellular carcinoma, SNRNP70, Immune infiltration, Bioinformatics, Prognosis

Funding: The National Natural Science Foundation of China Youth Training Project 2021GZR003 Medical-engineering Interdisciplinary Research Youth Training Project 2022YGJC001 This work was supported by the National Natural Science Foundation of China Youth Training Project (2021GZR003); Medical-engineering Interdisciplinary Research Youth Training Project (2022YGJC001). The funders had no role in study design, data collection and analysis, decision to publish, or preparation of the manuscript.

==============================
Background & Aims

Small nuclear ribonucleoprotein U1 subunit 70 (SNRNP70) as one of the components of the U1 small nuclear ribonucleoprotein (snRNP) is rarely reported in cancers. This study aims to estimate the application potential of SNRNP70 in hepatocellular carcinoma (HCC) clinical practice.

Methods

Based on the TCGA database and cohort of HCC patients, we investigated the expression patterns and prognostic value of SNRNP70 in HCC. Then, the combination of SNRNP70 and alpha-fetoprotein (AFP) in 278 HCC cases was analyzed. Next, western blotting and immunohistochemistry were used to detect the expression of SNRNP70 in nucleus and cytoplasm. Finally, Cell Counting Kit-8 (CCK-8) and scratch wound healing assays were used to detect the effect of SNRNP70 on the proliferation and migration of HCC cells.

Results

SNRNP70 was highly expressed in HCC. Its expression was increasingly high during the progression of HCC and was positively related to immune infiltration cells. Higher SNRNP70 expression indicated a poor outcome of HCC patients. In addition, nuclear SNRNP70/AFP combination could be a prognostic biomarker for overall survival and recurrence. Cell experiments confirmed that knockdown of SNRNP70 inhibited the proliferation and migration of HCC cells.

Conclusion

SNRNP70 may be a new biomarker for HCC progression and HCC diagnosis as well as prognosis. SNRNP70 combined with serum AFP may indicate the prognosis and recurrence status of HCC patients after operation.

Introduction

Liver cancer is composed of two major histology types, namely hepatocellular carcinoma (HCC) and cholangiocarcinoma (Chaisaingmongkol et al., 2017). Among them, HCC is the main histological subtype with a poor prognosis (Mulé et al., 2020), and the incidence of HCC worldwide increases by about 700,000 people every year, whose distribution depends on ethnicity, geographic region and so on (Petruzziello, 2018; Yang et al., 2018). Commonly, HCC is led by hepatitis B virus or hepatitis C virus, less correlated with sexual diseases, metabolic diseases, and genetic diseases (Tang et al., 2018; Schütte, Bornschein & Malfertheiner, 2009). However, aberrant molecular genes and signaling pathways in HCC tumorigenesis have gradually been recognized as important roles in the occurrence of HCC (Chen & Wang, 2015; Dimri & Satyanarayana, 2020). Thus, identifying molecular biomarkers in HCC is of great significance for the development of specific treatments for HCC.

The small nuclear ribonucleoprotein U1 subunit 70 (SNRNP70) is a well-known RNA-binding protein (RBP) primarily recognized for its nuclear function. SNRNP70 binds to U1 small nuclear RNA (U1 snRNA) and associates with other subunits to form the major spliceosomal component, U1 snRNP (Yin et al., 2020). SNRNP70 comprises a C-terminal tail domain that mediates interaction with Sm proteins, an α-helical domain, an RNA recognition motif (RRM) involved in contacting U1 snRNA, and two low-complexity arginine/serine domains facilitating interactions with SR proteins (Nikolaou et al., 2022). SNRNP70 is essential for both constitutive and alternative pre-mRNA splicing. Recent studies suggest its potential involvement in cytoplasmic processing of immature mRNAs (Nikolaou et al., 2022). The gene SNRNP70 is an encoding gene, and related to mixed connective tissue disease and facial atrophy (Hirobumi & Jun, 2016; Agnieszka et al., 2019). Recently, some studies show that SNRNP70 has a great potential in tumor treatment (Pierceall et al., 2011; Zhen et al., 2018; He et al., 2013), while the relation between SNRNP70 and HCC is still unclear.

To better understand the role of SNRNP70 gene in HCC, we used the publicly accessible databases (Travaglino et al., 2020; Beatty & Janelle, 2020) and functional experiments to perform SNRNP70 prognostic evaluation and clinicopathological significance. Through analyses, higher expression of SNRNP70 was found in HCC rather than peritumoral tissues, which might indicate poor overall survival (OS). Meanwhile, low expressions of nuclear SNRNP70 and alpha-fetoprotein (AFP) in HCC tissues had better OS prognosis, and low expression of SNRNP70 significantly inhibited the proliferation and migration of HCC cells. Therefore, we concluded that SNRNP70 could serve as a key biomarker for the prognosis of HCC.

Materials and Methods

TMT proteome analysis

We selected four paired frozen HCC and non-tumor liver tissues for proteomic profiling by Tandem mass tag (TMT, Thermo Fisher Scientific, Waltham, MA, USA). After protein extraction, trypsin digested the whole protein in tissue cells into peptides, which were then all tagged with TMT reagent. The labeled peptides from each sample were pooled and subjected to high-pH fractionation utilizing the Agilent 1,260 Infinity II High-Performance Liquid Chromatography (HPLC) system for separation. A total of 5 percent acetonitrile (ACN), 10 mm of HCOONH4, pH 10.0 served as buffer A, and 85 percent ACN, 10 mm of HCOONH4, pH 10.0 served as buffer B. With buffer A, the chromatographic column was brought to equilibrium. The manual injector fed samples into the chromatographic column for separation (1 mL/min flow). After being lyophilized, the eluting components were redissolved in 0.1 percent formic acid (FA). Thermo Fisher Scientific’s Easy nLC system was used to separate each sample, and the Orbitrap Exploris 480 mass spectrometer was used to conduct the analysis (Thermo Fisher Scientific, Waltham, MA, USA). The original atlas file created by the Orbitrap Exploris 480 was converted using the Proteome Discoverer 2.6 (Thermo Fisher Scientific, Waltham, MA, USA) software into a.mgf file, which was then sent to the MASCOT 2.6 server for database retrieval using the program’s built-in function. After that, the proteome discoverer 2.6 program returned the database search file (.dat file) created on the MASCOT server, and the data were filtered in accordance with the FDR < 0.01 criterion to produce qualitative findings. Herein, proteins were deemed to be differentially expressed proteins (DEPs, fold changes > 1.25 or <0.80, P < 0.05) (Jin et al., 2011). The enriched biological processes in Gene Ontology (GO) on DEPs were performed later.

Patients’ samples

We randomly collected formalin-fixed paraffin-embedded samples from 42 cases of HCC tissue with peritumoral tissue as the expression pattern cohort and 278 HCC tissue as the prognosis cohort. OS represents the interval from surgery to death or final examination. TTR referred to the time from tumor resection to recurrence, death, or final examination. Each patient involved in this study received abdominal ultrasound, serum AFP concentration and chest X-ray observation every 1 to 6 months during the first year after surgery, and every 3 to 6 months after that. Every 6 months or there was a suspected recurrence, we checked the abdominal CT scan or MRI. The same diagnostic criteria were strictly implemented. All samples were investigated by two senior liver pathologists. This study was approved by the Institutional Review Board at Eastern Hepatobiliary Surgery Hospital (EHBH; Approval number: EHBHKY2014-03-006). The informed consent was obtained from all patients.

Gene expression in pan-cancers

The expression data of SNRNP70 in different types of tumors were from TIMER (https://cistrome.shinyapps.io/timer/) (Li et al., 2017) and TCGA (https://tcga-data.nci.nih.gov/tcga) (Travaglino et al., 2020). The data were calibrated using R (version 3.3) to adjust the background, batch process, transform probes into gene symbols and standardize. The linear models of microarrays between 50 pairs of HCC and normal tissues were analyzed using the t-test program.

The correlation of SNRNP70 expression with HCC patients’ prognosis and clinical features

We used the Kaplan Meier plotter (http://kmplot.com/) (Zwyea, Naji & Almansouri, 2021) to access the effect of SNRNP70 expression on the survival rate of HCC. The UALCAN database (http://ualcan.path.uab.edu/index.html) (Chandrashekar et al., 2017) was for exploring the relationship between SNRNP70 and clinical characteristics.

Functional enrichment and protein-protein interaction analysis

Next, we used the Similar Genes Detection in GEPIA2 (http://gepia2.cancer-pku.cn/#similar) (Tang et al., 2019) to search for SNRNP70 co-expressed genes (1,000 in total) in HCC, displayed by protein-protein interaction (PPI) network. Then, the biological functions of 1000 genes were analyzed by Gene Ontology (GO) and Kyoto Encyclopedia of Genes and Genomes (KEGG).

Immune correlation analysis on SNRNP70

Then, the relation between SNRNP70 and immune infiltration was investigated. The TIMER database was applied to explore SNRNP70 and immune cell infiltration markers (B cell, CD8+ T cell, CD4+ T cell, Macrophage, Dendritic cell, Neutrophil, PDCD1, CD274). To describe the relation between SNRNP70 and tumor immune infiltration in detail, we set the following absolute values: 0.00–0.19 “very weak”, 0.20–0.39 “weak”, 0.40–0.59 “medium”, 0.60–0.79 “strong” and 0.80–1.0 “very strong”. Kaplan–Meier plotter was applied to assess the prognostic value of SNRNP70 in HCC based on immune infiltration.

Cell culture

The human HCC cell lines (pLC/RPF/5, Huh7, SK-Hep1) purchased from American Type Culture Collection (ATCC, Manassas, VA, USA). HEK-293T cells were cultured in RPMI-1640 medium with 10% FBS (fetal bovine serum, Gibco, Waltham, MA, USA), 100 U/mL penicillin, 100 mg/mL streptomycin (Gibco, Waltham, MA, USA), and other cell lines were cultured in DMEM medium (C11995500BT; Gibco, Waltham, MA, USA) with 10% FBS and two antibiotics. All cell lines were maintained in an incubator with 5% CO2 and 95% H2O at 37 °C.

Construction of CRISPR plasmid and virus gene transfection

Three double-stranded guide RNA (gRNA) sequences (gRNA-1, gRNA-2, gRNA-3) targeting of the SNRNP70 gene (NM_003089.6) were designed online using the web site (http://chopchop.cbu.uib.no), and oligonucleotide pairs for each gRNA were synthesized in Thermo Fisher company (the primers were demonstrated in Table S1). The pLenti CRISPR v2 plasmid was digested with BsmBI (NEB, Ipswich, MA, USA) and annealed to gRNAs. The lentiviral recombinant plasmid and control empty vector plasmid were produced using HEK293T cells according to Zhang Lab’s protocol (Shalem et al., 2014). Subsequently, virus containing recombinant plasmid and equal volume of medium with 10 ug/ml polybrene was added to the SK-Hep1 cells. After cells was cultured for three days, the puromycin with final concentration of 2 ug/ml was added to the medium for screening, and the surviving cells were sampled. Then the part of surviving cells was collected and subjected to Western blot analysis.

Western blotting and immunohistochemistry assays

Western blotting (WB) was performed for evaluating the protein expression in HCC and normal liver tissues, according to the immunohistochemistry and average optical density (AOD) measurement method. The polyclonal rabbit anti-human SNRNP70 antibody (WB 1:1000; IHC 1:500; ab83306; Abcam, Cambridge, UK) and anti-beta Actin antibody (WB 1:1000; Abcam; ab8226) was the primary antibody, and the secondary antibodies were goat anti-rabbit antibody (WB 1:5000; IHC 1:500; ab6721; Abcam, Cambridge, UK). The Envision detection kit allowed direct visualization of SNRNP70, and DAB as a chromogenic agent. We stained tissue sections for 5 min with hematoxylin. Negative control slides devoid of primary antibodies were prepared for each test. The NanoZoomer S60 was coupled to HALO software and a high-speed scanner for OD measuring.

Cell proliferation and cell migration assays

Cell proliferation assay were performed using Cell Counting Kit-8 (CCK-8, Abcam; Abcam, Cambridge, UK) method. According to the manufacturer’s instructions, 3,000 cells per well were placed into 96-well plates. Then, cells were treated with 10ul CCK-8 solution and incubated for 1 h at incubator. Optical density was measured at 450 nm wavelength. The testing was conducted every 24 h. To check if SNRNP70 played a role in cell migration, scratch wound healing assay was performed in SK-Hep1 cells. 4  × 105 cells were plated into 12-well plates. After cells was cultured for 24 h, a scratch was made using a pipette’s tip. Subsequently, wound-healing images were taken every 24 h to monitor the migration distances of cells.

Statistical analysis

X-tile software (Camp, Dolled-Filhart & Rimm, 2004) was applied to obtain the most suitable SNRNP70 expression cut-off point for survival analysis (Li et al., 2016; Wang et al., 2011). Mantel-Cox log-rank test was to assess the significance between SNRNP70 and patients’ survival in statistics. Besides, SPSS statistical software package was adopted to carry out a correlation analysis between variables. Independent sample t-test was used to compare differences between the two groups. The difference was statistically significant at P < 0.05.

Results

The results of TMT proteomics analysis

A total of 4,416 proteins were identified by TMT proteomics analysis from four pairs HCC and paracancer liver tissues, including 555 DEPs (FC > 1.25 or < 0.80 and P < 0.05, Fig. 1A). GO analysis on DEPs showed that these DEPs were closely related to mRNA splicing, via spliceosome, mitochondrial electron transport, steroid metabolic process and so on (Fig. 1B). SnRNP70 is a key complex for RNA splicing and spliceosome (Will & Lührmann, 2011). We focused on SNRNP70 for the next TCGA database analysis. Taken together, 555 DEPs were identified by TMT proteomics analysis and SNRNP70 may be associated with spliceosome in HCC.

Figure 1 The results of TMT proteomics analysis.

(A) The volcano plots of 4,416 proteins analyzed by TMT proteomics analysis.The x-axis represents the log2Fold_Change value and the y-axis represents log10 p_value. Red represents the up-regulated genes, while blue represents the downregulated genes. Gray represents no significant changes. (B) The enriched biological process analysis on 555 DEPs. The “count” on the right of the figure represents the number of genes enriched in the GO enrichment analysis corresponding to the GO term. Y axis indicates the GO term and X axis indicates GO enrichment p-value.

The expression of SNRNP70 in pan-cancers and its relationship with the prognosis and clinical characteristics of HCC

To study the role of SNRNP70 in the tumorigenesis of HCC, we investigated SNRNP70 levels in various tumors. As shown in Fig. 2A, the data in Oncomine and TIMER indicated SNRNP70 was highly expressed in most tumors, and the expressions of SNRNP70 in HCC tissues was statistically up-regulated (Fig. 2B). These results proved that SNRNP70 was a gene differentially expressed between HCC and normal samples. Next, the relation between SNRNP70 level, the prognosis and clinical characteristics was studied. Based on the median transcription level of SNRNP70, HCC patients were classified into two groups (high and low SNRNP70 groups). It was discovered the survival rate of high SNRNP70 group was lower than that of low SNRNP70 group (Figs. 2C–2F). Further, we studied the relation between SNRNP70 level and clinical characteristics in HCC. It could be seen from Fig. 3A that SNRNP70 had a higher expression in HCC tissues. The levels of SNRNP70 in advanced stages and high grades were higher than those in early stage and low grades (Figs. 3B and 3C). Based on nodal metastasis status, SNRNP70 level was higher in N1 stage rather than N0 stage (Fig. 3D). These findings demonstrated that high SNRNP70 levels were correlate with poor prognosis, advanced stages, high grades, and nodal metastasis in HCC.

Figure 2 SNRNP70 expressions levels in pan-cancers and Kaplan–Meier analysis on SNRNP70 in HCC.

(A) Analysis of SNRNP70 expressions levels in different types of tumor and normal tissues. ACC, Adrenocortical carcinoma; BLCA, Bladder urothelial carcinoma; BRCA, Breast invasive carcinoma; CESC, Cervical squamous cell carcinoma and endocervical adenocarcinoma; CHOL, Cholangio carcinoma; COAD, Colon adenocarcinoma; DLBC, Lymphoid Neoplasm Diffuse Large B-cell Lymphoma; ESCA, Esophageal carcinoma; GBM, Glioblastoma multiforme; HNSC, Head and Neck squamous cell carcinoma; KICH, Kidney Chromophobe; KIRC, Kidney renal clear cell carcinoma; KIRP, Kidney renal papillary cell carcinoma; LAML, Acute Myeloid Leukemia; LGG, Brain Lower Grade Glioma; LIHC, Liver hepatocellular carcinoma; LUAD, Lung adenocarcinoma; LUSC, Lung squamous cell carcinoma; MESO, Mesothelioma; OV, Ovarian serous cystadenocarcinoma; PAAD, Pancreatic adenocarcinoma; PCPG, Pheochromocytoma and Paraganglioma; PRAD, Prostate adenocarcinoma; READ, Rectum adenocarcinoma; SARC, Sarcoma; SKCM, Skin Cutaneous Melanoma; STAD, Stomach adenocarcinoma; TGCT, Testicular Germ Cell Tumors; THCA, Thyroid carcinoma; THYM, Thymoma; UCEC, Uterine Corpus Endometrial Carcinoma; UCS, Uterine Carcinosarcoma; UVM, Uveal Melanoma. (B) The expression levels of SNRNP70 mRNA in 50 pair of HCC and normal tissues. ***P < 0.001. (C) OS, the interval from surgery to death or final examination. The solid line shows the overall survival and the dotted lines shows the 95% confidence interval. (D) PFS, the interval from surgery to death or final examination.Number at risk indicates the number of patients at risk in each group at the corresponding time. (E) DFS, the interval from surgery to death or final examination. The solid line shows the disease free survival and the dotted lines shows the 95% confidence interval. (F) RFS, the interval from surgery to death or final examination. Number at risk indicates the number of patients at risk in each group at the corresponding time.

Figure 3 The relation between SNRNP70 and clinicopathological factors.

The level of SNRNP70 in hepatocellular carcinoma with different (A) sample types, (B) cancer stages, (C) tumor grade, (D) nodal metastasis status is based on the TCGA data. *P < 0.05, ***P < 0.001.

PPI and functional annotation analysis on SNRNP70 co-expressed genes

Next, we screened SNRNP70 co-expressed genes for the following study, and the PPI network in Fig. 4A contained 114 nodes and 308 edges in. In Fig. 4B, SNRNP70-related genes were mainly enriched in mRNA processing and RNA processing through the biological process (BP), and in nucleus and intracellular membrane-bounded organelle through cellular component (CC), respectively. Through molecular functions (MF), genes were enriched in RNA binding, and other terms. The enrichment of the KEGG pathway on SNRNP70-related genes indicated that spliceosome was the most enriched pathway (Fig. 4C). Taken together, these findings highlight the possible relevance of SNRNP70 to the spliceosome during HCC tumourigenesis and progression.

Figure 4 PPI network and functional analysis on SNRNP70 co-expressed genes.

(A) PPI network of SNRNP70 co-expressed genes with 114 nodes and 308 edges. V shapes represent genes with degrees between 0 and 5, ellipses represent genes with degrees between 6 and 15, and diamond shapes represent genes with degrees between 16 and 30. (B) GO annotations of SNRNP70 and similar genes in HCC. (C) The KEGG pathway of SNRNP70 co-expressed genes.

The immune correlation analysis on SNRNP70

It has been known there are significant differences in the immune scores of HCC, and different tumor infiltrating immune cells represent different prognosis of HCC (Meng et al., 2021). Thus, we analyzed the immune correlation of SNRNP70 in HCC. We found that SNRNP70 was positively correlated with B Cell (partial. cor = 0.403, P = 7.65e−15), CD8+ T Cell (partial. cor = 0.222, P = 3.43e−05), CD4+ T Cell (partial. cor = 0.388, P = 7.87e−14), Macrophage (partial. cor =0.375, P = 8.23e−13), Neutrophil (partial. cor = 0.33, P = 3.33e−10) and Dendritic Cell (partial. cor =0.327, P = 6.48e−10, Fig. 5A). SNRNP70 was positively correlated with natural killer cell specific markers, such as KIT (partial. cor = 0.244, P = 1.90e−06), ITGB2l (partial. cor = 0.246, P = 1.78e−06), CSF2 (partial. cor = 0.292, P = 9.72e−09), CXCR3 (partial. cor = 0.216, P = 2.80e−05), CD56 (partial. cor = 0.24, P = 2.93e−06) B3GAT1 (partial. cor = 0.26, P = 3.79e−07) and KLRC2 (partial. cor = 0.234, P = 5.44e−06, Fig. 5B). Taken together, these findings suggest that SNRNP70 may play an important role in immune escape in the liver cancer microenvironment.

Figure 5 SNRNP70 immune correlation analysis based on immune infiltration.

(A) The expression of SNRNP70 was positively correlated with the level of B cell, CD4+ T cell Macrophage, CD8+ T Cell, Neutrophil and Dendritic Cell immune infiltration levels of HCC. (B) SNRNP70 was positively correlated with natural killer cell specific markers, such as KIT, ITGB2l, CSF2, CXCR3, CD56, B3GAT1 and KLRC2. Black dots represent samples, the blue line represents the slope obtained through linear regression, and the grey shadows represent the 95% confidence interval.

Expression characteristics of SNRNP70 in HCC and peritumoral tissues

The results of WB showed the SNRNP70 protein expression in HCC tissues was higher (Figs. 6A–6B, P = 0.0065). Then, in Figs. 6C–6E, SNRNP70 staining of the specimen showed nuclear immunoreactivity of HCC and surrounding tissues, and the expressions of SNRNP70 in cell nuclear and cytoplasm were both higher in HCC tissues (Figs. 6F–6G). The OD value was then input into GraphPad Prism software. X-tile software was used to analyze 278 cases of HCC patients to determine the nuclear OD value expressed by SNRNP70 and the best cut-off point of cytoplasmic OD. We applied the standard log-rank method to select nuclear OD and cytoplasmic OD as the best cut-off points. Taken together, these findings highlight that SNRNP70 may have a close relationship with HCC, and play a promoting role in the process.

Figure 6 WB and the immunohistochemical experiments on SNRNP70.

(A, B) The protein levels of SNRNP70 in 10 paired HCC tissues and adjacent liver tissues (Adj). Actin was the reference gene. (C–E) Immunohistochemical images of SNRNP70. (F–G) Box plots showed the nuclear and cytoplasmic OD of SNRNP70 gene in 42 paired HCC and Adj tissues.

Correlation between SNRNP70 expression and clinicopathological characteristics

Two hundred and seventy-eight patients, including 34 women (12%) and 244 men (88%) was used as prognostic cohort. The assessed patient clinicopathological characteristics included sex, age, HBsAg, serum AFP, liver cirrhosis, TNM, Child-Pugh class, tumor size, tumor differentiation and vascular invasion. The clinical data and clinicopathological characteristics of the prognostic cohort were demonstrated in Table S2. It was found a correlation between the expression of SNRNP70, a key point on the basis of result classification, and clinicopathological factors in 278 HCC patients. Research indicates that combining immunohistochemical markers and serum markers can significantly enhance prognostic value compared to using them individually (Jin et al., 2013). Around 70% of HCC patients experience elevated serum levels of AFP. AFP is a key diagnostic and prognostic marker for HCC and serves as an immunotherapeutic target as a tumor-associated antigen in liver cancer (Zhao et al., 2022; Wang & Wang, 2018). In our investigation, we have identified a significant correlation between the expression of SNRNP70 and AFP levels (P = 0.032). Overall, these findings suggest that SNRNP70 may participate in HCC development by regulating serum AFP levels.

The results of Kaplan–Meier analysis

The X-tile program was chosen to acquire the best cut-off point of SNRNP70 expression for prognostic cohort survival analysis. As a result, 278 cases of HCC were divided into two groups in the analysis, with 139 cases in the low NOD group and 139 cases in the high NOD group. The results demonstrated the average OS time of HCC patients with low-NOD SNRNP70 value was higher than that of high NOD SNRNP70 value (Fig. 7A). The average TTR time of HCC patients with low NOD SNRNP70 expression was lower than that of high NOD SNRNP70 expression (Fig. 7B). Then, 278 cases of HCC were also divided into two groups in the SNRNP70 expression analysis in the cytoplasm, with 211 cases in the low cytoplasmic OD group and 67 cases in the high cytoplasmic OD group. The average OS time of HCC patients with low cytoplasmic OD SNRNP70 value was higher than that of high cytoplasmic OD SNRNP70 value (P = 0.057, Fig. 7C). The average TTR time of HCC patients with low cytoplasmic OD expression was lower than that of high cytoplasmic OD expression (P = 0.184, Fig. 7D). Finally, 278 HCC cases were divided into three groups when analyzing the combination of SNRNP70 and AFP in HCC prognosis, including 57 cases of low SNRNP70 nuclear and low AFP, 122 cases of low SNRNP70 nuclear and high AFP or high SNRNP70 nuclear and low AFP, 99 cases of high SNRNP70 nuclear and high AFP. The results showed that the OS time of 57 cases was higher than those of the other two groups (Fig. 7E), and the TTR time of 99 cases was higher than that of the other two groups (Fig. 7F). These results suggest that combination of low SNRNP70 nuclear and low AFP levels was associated with the highest overall survival time and the low relapse time in HCC patients.

Figure 7 Combined prognostic analysis of SNRNP70 nuclear expression and AFP in HCC patients.

(A–B) The average OS (P=0.036) and TTR (P = 0.184) time of 278 HCC patients in low and high NOD SNRNP70 groups. (C–D) The average OS (P = 0.057) and TTR (P = 0.184) time of 278 HCC patients in low and high cytoplasmic OD SNRNP70 groups. (E) SNRNP70 nuclear low and AFP low have the best prognostic effect in the nucleus of OS (n = 57). (F) SNRNP70 nuclear high and AFP high have the best prognostic effect of TTR in the nucleus (n=99)

Univariate and multivariate survival analysis

Univariate analysis demonstrated there existed a significant correlation between liver cirrhosis, tumor number, tumor size and nuclear SNRNP70/AFP combination and OS (P value for liver cirrhosis = 0.001, tumor size < 0.0001, tumor number = 0.001 and nuclear SNRNP70/AFP combination = 0.003). Meanwhile, serum AFP, whether there was liver cirrhosis, tumor size, and number of tumors were significantly correlated with TTR (serum AFP value = 0.044, P value for liver cirrhosis = 0.002, tumor size < 0.0001, tumor number < 0.0001, Table S3). The multivariate analysis adopted the Cox multivariate proportional hazard regression model and was conducted step by step (positive, conditional likelihood ratio). The results we obtained indicated that serum AFP, tumor size, cirrhosis, tumor number and SNRNP70 NOD were valuable prognostic indicators for OS and TTR. In addition, nuclear SNRNP70/AFP combination was an independent prognostic factor for overall survival but not for time to recurrence (Table S3). These results suggest that nuclear SNRNP70/AFP may serve as an independent prognostic biomarker for HCC.

Knock-down SNRNP70 in HCC cells inhibits HCC cell proliferation and migration

WB was performed to detect SNRNP70 protein expression level in HCC cell lines. The results showed that SK-Hep1 expression in protein level was higher than other HCC cell lines (Fig. 8A). Therefore, we knocked down SNRNP70 in SK-Hep1 cell and detected the SNRNP70 expression level by WB (Fig. 8B). To assess SNRNP70 function in HCC cells, we carried out cell proliferation with CCK-8 and cell migration with scratch wound healing assay. The downregulation of SNRNP70 inhibited cells growth in SK-Hep1 cells (Fig. 8C). Moreover, downregulation of SNRNP70 exhibited significantly slower repair of wound as compared to control group by scratch wound healing assays (Fig. 8D). Taken together, these findings collectively suggest that SNRNP70 exerts a stimulatory influence on cellular proliferation and migratory capacity within HCC cells.

Figure 8 SNRNP70 inhibits proliferation and migration of HCC cells.

(A) WB detection of SNRNP70 protein expression levels in HCC cell lines. (B) The protein expression level of SNRNP70 in SK-Hep1 cell line was detected by WB. (C) Down-regulated SNRNP70 in CCK-8 experiments inhibited the proliferation of SK-Hep1 cells. (D) The effect of down-regulation of SNRNP70 on HCC cell migration was measured by the scratch wound healing assay. **P < 0.01.

Discussion

The symptoms of patients with early HCC are not obvious (Ge & Huang, 2015), while patients with advanced HCC are often accompanied by liver pain, abdominal distension, anorexia, fatigue, weight loss, upper abdominal mass and so on (Draghici et al., 2019; Kumar & Panda, 2014). Currently, HCC is not sensitive to chemotherapy and radiotherapy, and the commonly-used treatment methods are liver transplantation and surgical resection (Rasool et al., 2014; Sala et al., 2004). Nevertheless, the recurrence rate of HCC patients is relatively high (Herrero et al., 2008), and metastasis rate is as high as 40%–70% in 5 years after surgical resection. Thus, it is still important to explore new methods in HCC diagnosis, treatment and prognosis. As high-throughput biotechnology develops, it has been possible to explore new treatment methods of HCC by identifying abnormally expressed genes, and reveal its molecular function and biological processes, which may help to discover valuable biomarkers for HCC diagnosis and therapy.

SNRNP70 is a component of the spliceosome U1 snRNP, which is essential for recognizing the 5′ splice site of the mRNA precursor and subsequent assembly of the spliceosome (Fan et al., 2021; Zahler et al., 2018). Current research reports chiefly pay attention to the spliceosome function of SNRNP70. For example, in the study of Mohammad Alinoor Rahman et al., the separation of the early spliceosome complex indicated the mutation weakened the binding of U1-70K (snRNP70) to the downstream 5′ splice site (Rahman et al., 2015). Interestingly, Smallwood et al. (2012) found that the recruitment of SNRNP70 and SRSF1 to IL1A was reduced when CBX3 was missing. There is no relevant report on the expression of SNRNP70 in HCC presently.

Similarly, there is currently no relevant report describing the prognostic effect of SNRNP70 in HCC and other cancers. Herein, we found the expressions of SNRNP70 were higher in tumor tissues and cell lines, and SNRNP70 knockdown could suppress cell proliferation in HCC. From our research, in general, SNRNP70 contributed to a poor prognosis in the progression of HCC. The four survival analysis charts indicated the high-expressed SNRNP70 had a lower survival time. In order to verify this result, we used the Kaplan–Meier tool to draw a survival analysis chart in the nuclear expression levels of SNRNP70 in 278 HCC patients. The results indicated the average OS time in low NOD SNRNP70 value group was higher than that of high NOD SNRNP70 value group, while the average TTR time in low NOD SNRNP70 expression group was lower. The average OS time in low cytoplasmic OD SNRNP70 value group was higher than that in high cytoplasmic OD SNRNP70 value, while the average TTR time in low cytoplasmic OD expression group was lower. These results indicate the high level of SNRNP70 nuclear results in the poor prognosis of HCC. However, in the combined prognostic analysis of 278 HCC patients with SNRNP70 and AFP, 57 patients with low SNRNP70 nuclear and low AFP had higher OS time than the other two groups, and 99 patients with high SNRNP70 nuclear and high AFP had longer TTR time than the other two groups. Thus, we come to a conclusion that the nuclear level of SNRNP70 is linked with the prognosis of HCC patients, and SNRNP70 combined with serum AFP could be a independent OS factor for HCC patients after operation.

To understand the mechanism of SNRNP70 in HCC, we explored the relationship between SNRNP70 level and the clinical factors of HCC. In the SNRNP70 expressions between tumor and normal tissues, SNRNP70 was highly expressed in tumor tissues. In the aspect of tumor staging, the level of SNRNP70 in HCC was higher in the higher stage than that in the lower stage. As for tumor grade, SNRNP70 expression in higher grades was higher than that in lower grades. Regarding nodal metastasis status, the level of SNRNP70 was higher in N1 stage than that in N0 stage. All these findings indicate that the level of SNRNP70 is highly expressed in various clinicopathological factors of tumors, and SNRNP70 could a new biomarker in HCC diagnosis and prognosis.

Besides, functional enrichment analysis exhibited SNRNP70 co-expressed genes were main linked with mRNA splicing, RNA splicing. More and more pieces of evidence show that splicing mechanisms are ubiquitous in various cancers and drive the generation and maintenance of cancer markers (Du et al., 2021). At present, studies have provided direct evidence on the carcinogenic potential of HCC-related splicing variants and their mechanisms. For example, Saito et al. (2002) reported an interesting case about the overexpression of DNA methyltransferase 3b (DNMT3b) splice variant DNMT3b4 in liver tissues from chronic hepatitis and cirrhosis and HCC tissue samples. The study by López-Cánovas et al. (2021) also showed that the overexpression of SF3B1 had been linked to HCC aggressiveness and survival. N6-methyladenosine (m6A) is a common mRNA nucleotide alteration that regulates mRNA stability, splicing, and translation (Chen & Wong, 2020). The work of Lin et al. (2020) revealed that RNA m6A methylation regulated sorafenib resistance in HCC by FOXO3-mediated autophagy. Korgaonkar et al. (2005) also showed that nucleophosprotein (NPM, also known as B23) was mainly located in the nucleolus and overexpressed in many types of human cancers. Then, the research results of Xu et al. (2014) showed that NPM protein is overexpressed in HCC cells and clarified that NPM played a regulatory role in HCC, and it was worthy of in-depth study.

Immune cells have been linked to the incidence, development, and prognosis of malignancies in a variety of studies (Zamarron & Chen, 2011). According to recent studies, several genes can influence the tumor microenvironment via immune cells (Zhu et al., 2021). As a result, we studied the connection between SNRNP70 and HCC immune cell infiltration. In this study, we placed SNRNP70 in HCC tissues for immune infiltration to explore the regulation of SNRNP70 on the tumor microenvironment. It was found the level of SNRNP70 was positively related to immune cells in tumor tissues. Cell experiments also confirmed that down-regulation of SNRNP70 expression significantly inhibited the proliferation and migration of HCC cells. In addition, in normal liver cells (THCE3), our observations revealed low expression levels of SNRNP70. Following the targeted knockdown of SNRNP70 in these cells, we noted a marked reduction in both the proliferation and migration of normal hepatic cells. This finding not only highlights that SNRNP70 may be an important target for HCC and its specific role in HCC pathogenesis, but also highlights its potential impact on the physiology of non-cancerous liver tissue. Herein, we analyze the clinical value of SNRNP70 in HCC progression, and find that the SNRNP70 is closely linked to the prognosis and clinical characteristics of HCC patients. These findings indicate SNRNP70 may be a potential biomarker for HCC diagnosis, treatment, and prognosis. When paired with serum AFP, SNRNP70 may also be able to predict postoperative results and the likelihood of tumor recurrence. However, further study is still required to explore the potential mechanism of SNRNP70 in HCC.

Supplemental Information

Supplemental Information 1 Sequences of the three pairs of gRNAs designed online

Supplemental Information 2 The relation between SNRNP70 expression and clinical factors in 278 HCC patients

Supplemental Information 3 Univariate and multivariate analysis of OS and TTR-related factors in 278 HCC patients

Supplemental Information 4 SNRNP70 inhibits the proliferation and migration of THCE3 cells

Abbreviations

HCC Hepatocellular carcinoma

SNRNP70 Small nuclear ribonucleoprotein U1 subunit 70

OS Overall survival

AFP Alpha-Fetoprotein

TMT Tandem Mass Tag

DEPs Differentially Expressed Proteins

GO Gene Ontology

EHBH Eastern Hepatobiliary Surgery Hospital

KEGG Kyoto Encyclopedia of Genes and Genomes

ATCC American Type Culture Collection

FBS Fetal bovine serum

gRNA guide RNA

WB Western Blotting

AOD Average optical density

CCK-8 Cell Counting Kit-8

BP Biological process

CC Cellular component

MF Molecular functions

DNMT3b DNA methyltransferase 3b

m6A N6-methyladenosine

NPM Nucleophosprotein

RBP RNA-binding protein

RRM RNA recognition motif

NOD Nuclear optical density

Additional Information and Declarations

Competing Interests

Author Contributions

Data Availability

The authors declare there are no competing interests.

Dong Jiang performed the experiments, analyzed the data, prepared figures and/or tables, authored or reviewed drafts of the article, methodology; Investigation, and approved the final draft.

Xia-Ling Zhu analyzed the data, authored or reviewed drafts of the article, and approved the final draft.

Yan An conceived and designed the experiments, authored or reviewed drafts of the article, and approved the final draft.

Yi-ran Li conceived and designed the experiments, performed the experiments, analyzed the data, prepared figures and/or tables, authored or reviewed drafts of the article, and approved the final draft.

The following information was supplied regarding data availability:

The raw data is available at figshare and Zenodo:

- Li, Yiran (2024). Untitled Item. figshare. Dataset. https://doi.org/10.6084/m9.figshare.25239364.v1.

- Li, Y. (2023). Raw Data. Zenodo. https://doi.org/10.5281/zenodo.10299097.

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
