# Peer review of "Clinical significance of small nuclear ribonucleoprotein U1 subunit 70 in patients with hepatocellular carcinoma"

_PeerJ, doi:10.7717/peerj.16876_

## Round 0.1 · original submission · Major Revisions

The manuscript needs substantial revision, additional work and justifications in order to appreciate the quality for publication. Reviewers have commented against the acceptance of the manuscript in its current form and it suffers from serious concerns regarding the implemented protocol as well as presentation of the data. Moreover, thorough English editing is required. Please revise the manuscript taking help from a colleague who is proficient in English and familiar with the subject matter, who can review your manuscript, or contact a professional editing service to review your manuscript. Revise and resubmit accordingly.

·

Basic reporting

Regarding the manuscript, I would like to bring to your attention a few issues with the figures and their descriptions:

1. The Figure 1 description and Figure 2 description on page 35 and page 36 should be interchanged. Currently, Figure 1 is described as "SNRNP70 expressions in pan-cancers and Kaplan-Meier analysis on SNRNP70 in HCC," while Figure 2 is described as "The results of TMT proteomics analysis," which is inconsistent with the figures presented and the results described in your manuscript.

2. The Figure 3 description on page 38 should read "hepatocellular carcinoma" or "liver cancers" instead of "lung cancers."

3. The text font of Figures 4B, 4C, and 5 may need to be enlarged to enhance readability, as it is not very clear at present.

4. Statistical significance (P value) should be added to Figures 6F and 6G.

5. In Figure 8D, "SNRNP70-KD" should be used instead of "SNRNP70."

Experimental design

1. In Figures 7E and 7F, I noticed that you combined SNRNP70 with AFP for HCC prognosis. Could you please provide some background information on why you chose AFP over other proteins? It would be helpful to add some justification for this selection.

2. In Figure 7, there is an abbreviation "NOD," but its meaning is not provided. It would be helpful to add NOD and its meaning to the "Abbreviation" section on page 7.

3. How did you measure SNRNP70 expression in Figure 4?

Validity of the findings

no comment

Reviewer 2 ·

Basic reporting

Fail. (See Additional comments)

Experimental design

Fail. (See Additional comments)

Validity of the findings

Fail. (See Additional comments)

Additional comments

Dear Editor,

I have carefully reviewed the manuscript entitled “Clinical significance of small nuclear nucleoprotein U1 subunit 70 in patients with hepatocellular carcinoma” submitted by Dong Jiang et al. While the study topic is of interest, there are some major concerns that lead me to recommend rejection of the manuscript in its current form.

The manuscript explores the potential application of SNRNP70 in hepatocellular carcinoma (HCC) clinical practice. The authors demonstrate that SNRNP70 is highly expressed in HCC and related to immune infiltration cells. They also suggest that higher SNRNP70 expression is associated with poor patient outcomes and that the combination of nuclear SNRNP70 and alpha-fetoprotein (AFP) could serve as a prognostic biomarker for overall survival and time to recurrence. However, there are major concerns regarding the number of patients included in various analytical assays and the classification of SNRNP70 as an oncogene. Addressing these concerns and providing additional experiments to support the claims made in the manuscript are necessary before considering this work for publication on PeerJ.

Major Concerns:
1. The number of HCC patients included in various analytical assays is unclear. The authors mention that this study was performed on 278 HCC patients. However, the TMT label-based proteomic profiling experiment was only performed on 4 paired frozen HCC and non-tumor liver tissues; it appeared the Western blotting assay was only performed on 10 HCC patients’ samples, and the immunohistochemistry assay was performed only on one patient’s sample as indicated in Figure 6. It is also confusing that the “Gene expression in pan-cancer” experiment reads like a meta-analysis using a public database. To strengthen the manuscript, the authors should collect SNRNP70 differential expression data from 278 HCC patients’ samples and conduct a multi-omics study at genomics, transcriptomics, and proteomics levels to support their hypothesis that “SNRNP70 is a new biomarker highly expressed in HCC.”

2. Classifying SNRNP70 as an “oncogene in HCC progression” (Line 42) is inappropriate, as SNRNP70 is a housekeeping gene involved in pre-mRNA splicing, responsible for maintaining basic cellular functions and generally expressed in all cell types. The SNRNP70 knock-down experiment is not plausible because if a housekeeping gene is compromised, any type of cells would become less healthy and have limited growth. To prove this hypothesis, an additional control knock-down experiment on healthy liver cells would be needed to evaluate the housekeeping nature of SNRNP70.

Minor Concerns:
1. Line 87: “proteomics data” misspelled as “protemics data.”
2. Line 98: “publicly accessible databases” may need references.
3. Line 109: The term “proteomic characterization” is not common; “proteomic profiling” is the more suitable term here.
4. Line 110: Supplementary Table 1 appears to be the wrong table that is irrelevant to the proteomics experiment.
5. Line 111 and 112: There is no information on an indispensable total protein quantitation assay such as BCA or Bradford assay. If the authors did not include such assay in their study, the TMT based quantitation results might be less reliable because TMT labeling relies on NHS ester-amine reaction at peptide level, difference in protein sample input would significantly affect the final quantitation accuracy.
6. Line 112: It is not clear what the “ranked” means. The author apparently performed high-pH fractionation of TMT labeled peptides on the Agilent 1260 HPLC, which can be further clarified.
7. Line 113: Acronym “ACN" is not defined. I assume it means acetonitrile, but other readers may not be familiar with this acronym.
8. Line 115: The term “manual automated” is self-contradicting.
9. Line 117: Acronym “FA" is not defined. I assume it means formic acid, but other readers may not be familiar with this acronym.
10. Line 122: Which reference protein sequence database? Swiss-Prot human proteome database? It is essential to report the reference protein sequence database in any proteomic profiling experiment.
11. Line 122 and 123: It is not clear whether Proteome Discoverer 2.6 or Proteome Discoverer 2.2 or both were used in the data processing. Why is there a version difference?
12. Line 125: The term “extremely trustworthy” is an overstatement of some sort and may not be appropriate in its context.
13. Line 126: The DEP screening criteria, fold changes>1.25 or <0.80, P<0.05, seems arbitrary. The authors may need to provide references or add justification to their rationale for setting such criteria.
14. Line 144: This domain has been claimed by Thermo Fisher and redirects to https://www.oncomine.com/ as "Oncomine Solutions For NGS."
15. Line 146: It is not clear whether the authors performed microarray analysis of HCC and normal tissues from how many patients.
16. Line 163: Why are NK cell specific markers such as CD56 and CD16 missing on the immune cell infiltration marker panel?
17. Line 182: “Thermo Fisher company” misspelled as “Thermo Fischer compony.”
18. Line 218 to 223: The authors may want to report all 4,416 proteins identified in the proteomic profiling experiment. The author may also want to highlight some details on relative quantitation of SNRNP70 and AFP from the mass spectrometry data.
19. Line 234 and 235: It is unclear how high is “high”, how low is “low” here.
20. Figure 1: The figure description appears to mismatch the actual figure.

In conclusion, due to the major concerns and the significant revisions and additional experiments required to support the claims made by the authors, I recommend rejecting the manuscript in its current form. The authors are encouraged to address these concerns and provide more robust evidence before considering resubmission to ensure a stronger and well-supported study.

Sincerely,
PeerJ Reviewer

Reviewer 3 ·

Basic reporting

Please see Additional comments.

Experimental design

Please see Additional comments.

Validity of the findings

Please see Additional comments.

Additional comments

The authors combined bioinformatics and cell experiments to explore the role of SNRNP70 in hepatocellular carcinoma pathogenesis and prognosis. Overall, this study is suitable for publication, only if the authors address the following issues:

1. Throughout the manuscript, it seems better to use Grammarly (https://www.grammarly.com/) to check & correct potential grammatical errors or typos. For example,
1.1 In Figure 2's legend, it seems better to change "SNRNP70 expressions" into "SNRNP70 expression patterns" or "SNRNP70 expression levels".

2. In all FIGURES, it would be more clear and more readable to expand on figure legends by explaining the meanings of colors, groups, lines, and abbreviations. For example,
2.1 In the current manuscript, the Figure 1's title & legend and Figure 2's were mistakenly swapped.
2.2 In Figure 1A's legend, it seems better to mention what is the meaning of the dots' colors (red and blue) and "NS". (please see how all elements in a volcano plot were explained by this article: PMID_27518660)
2.3 In Figure 1B, it seems better to delete GO numbers ("GO: 0000398") of terms; in the figure legend, it seems better to explain the meaning of "Count" (the node size).
2.4 In Figure 2A's legend, it seems better to mention the full names of the cancer abbreviations on the Y axis. In Figures 2C & 2E's legends, it seems better to mention the meaning of dotted lines. In Figure 2D & 2F's legends, it seems better to mention the meaning of black & red numbers below the images.
2.5 In Figure 4A's legend, it seems better to mention the meaning of different shapes.
2.6 In Figure 5's legend, it seems better to mention the meaning of black dots, blue lines, and grey shadows.
2.7 In Figure 8D, it seems more accurate to change "SNRNP70" into "SNRNP70-KD".

These revisions would greatly help readers, who do not specialize in bioinformatics, to understand the results and their implications easily and efficiently.

3. In ABSTRACT:
3.1 In Background, it seems better to change "Small nuclear nucleoprotein U1 subunit 70" into "Small nuclear ribonucleoprotein U1 subunit 70", which is more widely used.
3.2 In Background, it seems better to change "snRNP" into "small nuclear ribonucleoprotein (snRNP)", whose full name would help readers understand better.
3.3 In Methods, it seems better to delete "SNRNP70 was focused in proteomics and GO analyses", which could not add something helpful for readers to understand.
3.3 In Methods, it seems better to change "The characteristics of the expressions and prognosis of SNRNP70 in HCC were collected from 278 HCC patients and explored using TCGA database" into "Based on the TCGA database, 278 HCC patients data were collected to explore the expression patterns and prognostic value of SNRNP70 in HCC", which would be clearer and parallel to sentences after it (i.e., "Finally, western blotting ... SNRNP70 protein, and Cell Counting Kit-8 (CCK-8) ... migration of HCC cells").
3.4 In Methods, it seems better to change "Finally, western blotting and immunohistochemistry were used to detect the expression of the nucleus and cytoplasmic SNRNP70 protein, and Cell Counting Kit-8 (CCK-8) and scratch wound healing assays were used to detect the effect of SNRNP70 on the proliferation and migration of HCC cells" into "Next, western blotting and immunohistochemistry were used to detect the expression of SNRNP70 in nucleus and cytoplasm. Finally, Cell Counting Kit-8 (CCK-8) and scratch wound healing assays were used to detect the effect of SNRNP70 on the proliferation and migration of HCC cells", which would be clearer.
3.5 In Results, it seems better to change "SNRNP70 was highly expressed in HCC and also increased with stepwise progression of HCC and was positively related to immune infiltration cells" into "SNRNP70 was highly expressed in HCC. Its expression was increasingly high during the progression of HCC and was positively related to immune infiltration cells", which would be clearer.
3.6 In Results, it seems better to change "In addition, nuclear SNRNP70/AFP combination could be a prognostic biomarker for overall survival and time to recurrence" into "In addition, nuclear SNRNP70/AFP combination could be a prognostic biomarker for overall survival and recurrence", which would be more concise.
3.7 In Conclusion, it seems better to change "SNRNP70 may be an oncogene in HCC progression, and it could be a new biomarker in HCC diagnosis and prognosis, and SNRNP70 combined with serum AFP may indicate the prognosis and recurrence status of HCC patients after operation" into "SNRNP70 may be an oncogene in HCC progression and a new biomarker in HCC diagnosis as well as prognosis. SNRNP70 combined with serum AFP may indicate the prognosis and recurrence status of HCC patients after operation", which would be clearer and have smoother logic flow.

4. In INTRODUCTION:
4.1 In Paragraph 1, it would be better to delete "Currently, researchers have paid more and more attention on HCC treatment. For example, research by Jie Zhang et al. demonstrates that CKS1 is a well-known cell cycle-related protein, which is related to various tumors, including breast cancer, lung cancer and HCC(7). Despite all that, HCC is still one of the most malignant tumors(8)", which did not seem necessary.
4.2 In Paragraph 2, it seems better to delete this section ("Recently, a growing number of platforms, databases and data sets have made ... Therefore, we focused on SNRNP70 for the next TCGA database analysis.") and replace it with a paragraph that introduces SNRNP70, which is this study's major character.
4.3 In Paragraph 3, it seems better to put "The gene SNRNP70 is an encoding gene, and related to mixed connective tissue disease and facial atrophy(11, 12). Recently, some studies show that SNRNP70 has a great potential in tumor treatment(13-15), while the relation between SNRNP70 and HCC is still unclear" in Paragraph 2 (a section introducing SNRNP70), because the introduction's final paragraph is usually a section that does not introduce research background but rather preview the current study.

5. In MATERIALS AND METHODS:
5.1 In "Western blotting (WB) and immunohistochemistry assays", it would be more rigorous to mention the antibodies' brand and lot number.
5.2 In "Statistical analysis", it would be more rigorous to mention the statistical methods (t-test/ANOVA?) and significance level (p < 0.05?) when comparing two/multiple groups of data points.

6. In RESULTS:
6.1 In "The results of TMT proteomics analysis", please explain why the authors focused on the proteins (including SNRNP70) that enriched for "mRNA splicing, via spliceosome", which seems a general (not specific) term — a biological process involved in not only tumorigenesis but also almost all homeostasis & disease. Likewise, please explain why the authors concentrated on SNRNP70 rather than other members of "the 37 proteins".
6.2 In "The expression of SNRNP70 in pan-cancers and its relationship with the prognosis and clinical characteristics of HCC", it would be clearer to rewrite "These findings demonstrated SNRNP70 level was indeed connected with the prognosis and clinicopathological factors of HCC" by replacing "clinicopathological factors" (which seems too general) with a more specific expression.
6.3 In "PPI and functional annotation analysis on SNRNP70 co-expressed genes", please explain why the authors conducted this analysis and how these results could advance findings in previous sections (in other words, what did these "PPI and functional annotation" results imply? How did these results relate to the hypothesis "SNRNP70 may be an oncogene in HCC progression, and it could be a new biomarker in HCC diagnosis and prognosis"?)
6.4 In "The immune correlation analysis on SNRNP70", it would be more rigorous to add references to support the statement "It has been known there are significant differences in the immune scores of HCC, and different tumor infiltrating immune cells represent different prognosis of HCC".
6.5 In "The immune correlation analysis on SNRNP70", please explain why the authors focused on "tumor purity", whose correlation with SNRNP70 seems very weak ("cor=0.064").
6.6 In "Expression characteristics of SNRNP70 in HCC and peritumoral tissues", please explain why the authors performed the following procedures "The OD value was then input into GraphPad Prism software. X-tile software was used to analyze 278 cases of HCC patients to determine the nuclear OD value expressed by SNRNP70 and the best cut-off point of cytoplasmic OD. We applied the standard log-rank method to select nuclear OD and cytoplasmic OD as the best cut-off points." If this is a widely used method, please cite references and expand on the method as well as its implications in MATERIALS AND METHODS. In addition, please present the data about "cut-off point" in Figure 6.
6.7 In "Correlation between SNRNP70 expression and clinicopathological characteristics", please mention the detailed "clinical data and clinicopathological characteristics" (i.e., what are the characteristics).
6.8 In "Correlation between SNRNP70 expression and clinicopathological characteristics", it would be more readable to mention why the authors paid attention to that "SNRNP70 expression was related to serum AFP".
6.9 It would be clearer to end each paragraph in RESULTS with one sentence: "Together, these results suggest that ..." (a pattern like PMID: 34715879, PMID: 34384362, PMID: 35965679, and PMID: 34537192), summarizing a paragraph AND highlighting the implications of all results in the paragraph.

7. In SUPPLEMENTAL FILES, it would be better to add "peerj-83880-approval_of_Ethics_Committee" which is in English, for the convenience of international readers.

---

## Round 0.2 · Major Revisions

Though the manuscript is significantly improved by the authors, reviewers still have raised some suggestions and concerns to improve the manuscript. Please revise considering the comments and resubmit. Concerns raised by the reviewers are extremely important and reflect the intended quality, and it is necessary to undertake and address all comments.

·

Basic reporting

I would still suggest enlarging the text font of Figures 4B, 4C, and 5. It is recommended that the font is as large as the font in Figure 1. If it is hard to adjust the font by the code, I suggest only copying the image to PowerPoint and manually adding the text.

All the other concerns from reviewer 1 have been resolved.

Experimental design

All the concerns from reviewer 1 have been resolved.

Validity of the findings

All the other concerns from reviewer 1 have been resolved.

Additional comments

All the concerns from reviewer 1 have been resolved.

Reviewer 2 ·

Basic reporting

The revised manuscript has made improvements in terms of clarity, formatting, and description. The alignment issue between Figure 1 and its description has been addressed.

Concerns:
1. The explanation regarding the classification of SNRNP70 has been altered, but further exploration on this aspect is still needed to make a strong contribution to the field.
2. A more detailed explanation of why SNRNP70 protein was one of the 37 proteins in the mRNA splicing, via spliceosome, but no relation was verified between SNRNP70 and tumor diagnosis and prognosis, is needed in the discussion section.

Experimental design

The design of the experiments has some unresolved issues that significantly impact the robustness of the study. Although improvements were made in describing the proteomic analysis, some aspects still remain unclear, requiring additional justification or referencing.

Concerns:
1. The patient sample size inconsistency remains a concern despite the explanation provided by the authors. A more substantial and consistent sample size would greatly enhance the scientific rigor.
2. The lack of control knock-down experiments, due to contamination concerns, leaves a significant gap in the experimental design.
3. What led to the inconclusive result concerning SNRNP70? Was it related to the selection of only 4 paired frozen HCC and non-tumor liver tissues? Was there any limitation in the experimental procedure that prevented conclusive findings?

Validity of the findings

The validity of the findings is still questionable due to the unresolved concerns. The validity of the proteomic results needs to be critically examined, with specific emphasis on the inconclusive result concerning SNRNP70.

Concerns:
1. Removing the characterization of SNRNP70 as an oncogene was a prudent decision, but the lack of further experiments to explore its role in HCC limits the validity of the findings.
2. Despite the authors' explanation about different sample sizes, concerns related to the robustness of the findings remain.
3. The manuscript must elucidate why SNRNP70 was identified in the context of mRNA splicing but could not be linked to tumor diagnosis and prognosis. Was it due to limitations in sample size, methodology, or something else?
4. How does this inconclusive finding affect the overall interpretation of the study, and what implications does it have for the understanding of SNRNP70's role in HCC?

Additional comments

I appreciate the authors' efforts to address my initial concerns, and I acknowledge the improvements made in the clarity and correctness of the manuscript. However, significant issues related to the classification of SNRNP70 and experimental design limitations remain unresolved.

Recommendations:
1. The manuscript requires substantial modifications to resolve the major issues mentioned above, especially the incorporation of further wet lab experimentation.
2. The manuscript may warrant reconsideration for publication only if these concerns are adequately addressed.

Conclusion:
While the revised manuscript shows efforts to address the concerns raised in the initial review, the remaining major issues prevent me from recommending publication in its current form. The clinical significance of SNRNP70 in HCC patients would require a more compelling case through a consistent and robust experimental design.

Reviewer 3 ·

Basic reporting

Thank the authors for their efforts to respond to all of my comments. However, a small part of comments have not been addressed thoroughly, so it would be better to deal with such issues:

1. As to the previous comment "2.3 In Figure 1B, it seems better to delete GO numbers ("GO: 0000398") of terms; in the figure legend, it seems better to explain the meaning of "Count" (the node size)", it would be clearer (easier to understand) to explicitly mention the word "Count", thereby explaining what is meant by "Count". The word "Count" was found in Figure 1B, so it would be necessary to explain what is the meaning of "Count" in Figure 1B's legend.

2. In the legend of Figure 1B, it seems better to change "Y axis indicates the GO term and X axis indicates GO enrichment value" into "Y axis indicates the GO term and X axis indicates GO enrichment p-value". After this revision, the sentence would be more accurate.

3. In Figure 1A, the red color did not seem to be noticeable, so it would be clearer (easier to read) to use a brighter shade of red.

4. As to the previous comment "6.5 In "The immune correlation analysis on SNRNP70", please explain why the authors focused on "tumor purity", whose correlation with SNRNP70 seems very weak ("cor=0.064")", the authors did not seem to respond as expected. To be clearer, it would be more informative to explain what is "tumor purity", how it was analyzed, why it is important, how its analysis has been used in other cancer studies (that is, reading and citing references), and how this analysis contribute to the current study's hypothesis (SNRNP70 plays a role in HCC). These aspects could be addressed both here and the manuscript's MATERIALS AND METHODS (the section "Immune correlation analysis on SNRNP70").

Experimental design

N/A

Validity of the findings

N/A

Additional comments

N/A

---

## Round 0.3 · accepted · Accept

Manuscript is significantly improved by the authors and now can be accepted in its current form.

·

Basic reporting

All concerns have been addressed. The manuscript is ready to publish.

Experimental design

All concerns have been addressed. The manuscript is ready to publish.

Validity of the findings

All concerns have been addressed. The manuscript is ready to publish.

Additional comments

All concerns have been addressed. The manuscript is ready to publish.

Reviewer 2 ·

Basic reporting

No more comments.

Experimental design

No more comments.

Validity of the findings

No more comments.

Additional comments

The authors have thoroughly explained and justified their responses to my earlier concerns. The decision regarding the publication of this manuscript is now at the editor's discretion.

Reviewer 3 ·

Basic reporting

Please see Additional Comments.

Experimental design

Please see Additional Comments.

Validity of the findings

Please see Additional Comments.

Additional comments

Thank the authors for responding to all of the comments. The current version has been improved.